# The Roles of Carbon Trading System and Sustainable Energy Strategies in Reducing Carbon Emissions—An Empirical Study in China with Panel Data

**DOI:** 10.3390/ijerph20085549

**Published:** 2023-04-17

**Authors:** Yue Yu, Yishuang Xu

**Affiliations:** 1Office of Academic Research, Central Academy of Fine Arts, Beijing 100000, China; 250271@network.rca.ac.uk; 2Manchester Urban Institute, Department of Planning and Environmental Management, University of Manchester, Manchester M13 9PL, UK

**Keywords:** carbon emissions, carbon emissions trading system (ETS), sustainable energy strategy, urbanization

## Abstract

Carbon emission reduction is now a vital element in urban development. This study explores the effectiveness of the two emerging methods to reduce carbon emission, which are carbon emissions trading system (ETS) and sustainable energy strategy, in the process of urbanization. We review the policy in the past decades to demonstrate the development of these two streams of carbon emission reduction methods and empirically test the effectiveness of the two methods with panel data across 30 provinces in China from 2009 to 2019. The sustainable energy strategy is confirmed to be effective in reducing carbon emissions in the region, while the effectiveness of carbon emissions trading system varies. We find that (1) substituting fossil fuel with other sustainable energy resources can effectively reduce the carbon emission; (2) the rewards from carbon emissions trading is a good incentive for the enterprises to reduce the carbon emissions; however, it is more tempting in the provinces that have the carbon emissions trading system, although the trading can be conducted across the province boarder. Our findings indicate that the sustainable energy strategy is a good practice and worth expanding to the whole country. It can be difficult for some provinces to transform and adopt the sustainable energy strategy if the fossil fuel is the major source for economic production. It is important to avoid setting fossil fuel as the main source for economic production or household consumption in the urbanization process. Meanwhile the carbon emissions trading system is found to contribute to CO_2_ emissions reduction only within the province. Therefore, having more provinces piloting the ETS will help the CO_2_ emission reduction further.

## 1. Introduction

Rapid urbanization may lead to high levels of carbon emissions, endangering the environment and human well-being. This can also exacerbate the geographical inequality. Studies show that a 1% increase in carbon emissions is associated with a 0.298% increase in outpatients and a 0.162% increase in inpatients [1]. Many countries have recognized the urgency of energy conservation and environmental protection and have attempted to employ various policies, mechanisms, and strategies. These include institutional and legal frameworks, such as government orders requiring high-energy-consuming companies to make corrections by a deadline, as well as economic instruments such as taxes and subsidies. In addition, countries have also leveraged technological progress and information to increase environmental awareness. To further address this issue, the carbon emissions trading system (ETS) has been adopted as a market-based incentivizing instrument. Under the ETS, a cap is established for greenhouse gases (or CO_2_) during a compliance period. Emissions allowances can be traded on the market as commodities [2]. Another measure to reduce the carbon emissions is to adopt the sustainable energy strategy by substituting the fossil fuel with other sustainable energy resources. In this study, we quantitatively investigate the effectiveness and the contribution of both measures in carbon emissions reduction in China.

In 2009, China pledged at the United Nations Climate Change Conference to reduce its CO_2_ emissions by 40–45% by 2020. The Chinese government has published a variety of strategies to improve resource efficiency and reduce carbon emissions. The China Carbon Emissions Trade Exchange (CCETE) includes seven pilot districts (2011), eight ETS pilots (2013), and a national ETS (2021). Seven provinces have ETS pilots, namely Beijing, Tianjin, Shanghai, Fujian, Hubei, Shenzhen, Guangzhou, and Chongqing. Guangdong province has both Shenzhen and Guangzhou ETS. These provinces are collectively known as CCETE provinces. Figure 1 demonstrates the carbon emissions across seven CCETE provinces in China from 2009 to 2019.

It shows that the CCETE provinces have made a significant contribution to carbon emissions reduction by maintaining the low levels of carbon emissions with a small growth rate, especially in comparison to other provinces without ETS pilot. Figure 2 shows the annual growth of carbon emissions in each province in China between 2009 and 2019. Whilst some provinces (i.e., Tianjin, Hebei, Jiangsu, Henan, Hunan, Guizhou, Yunnan, and Qinghai) showed a carbon emission reduction over time, some other provinces which heavily rely on the energy-intensive economic growth showed persisting high levels of carbon emissions. In these provinces, the financial incentives from ETS might not be adequate to trade-off the loss from the economic growth if they reduce energy use. In this case, the sustainable energy strategy is vital. By substituting the fossil fuel with the sustainable energy resources, the province economy will not suffer the loss while the carbon emissions can be reduced.

This study aims to examine the impact and effectiveness of the two measures, namely ETS and Sustainable Energy Strategy, in carbon emissions reduction. Panel data in China from 2009 to 2019 are employed in our empirical tests. Our findings will shed some light to the stakeholders, such as the policy makers and local authorities, regarding the future urbanization progress. 

In the following section, we will review the relevant literature in this area and briefly introduce the development of sustainability policies, including the construction of CCETE (pilot districts, ETS pilots, and national ETS). We follow the Kaya [3] model to develop our empirical model and collect the empirical data from public data sources. The results and findings will be discussed and followed by the conclusions with policy implications. 

## 2. Literature Review

### 2.1. What Affects Carbon Emissions?

Since the 1990s, factors affecting carbon emissions have gained wider attention, with the most famous example being the Kyoto Protocol proposed by the United Nations Framework Convention on Climate Change (UNFCCC) in 1997. Kaya presented the concept of the Kaya Identity based on IPAT [3,4]. This is widely accepted as an effective way to calculate carbon emissions and establish the relationship between carbon emissions, economic policies, population, and human activity. Later in 1993, Kaya co-authored with Yokoboriet and developed the Kaya Identity further [5]. The relationship can be calculated by global carbon emissions from human sources (F). The Kaya Identity has four components: total population (P), GDP per capita (G), energy intensity of GDP (E/G), and carbon intensity of energy (F/E). The relationship between these factors can be expressed in the Equation:(1)$$F_{CO2\;\mathrm{emissions}\mathrm{from}\mathrm{human}\mathrm{sources}}=P_{\mathrm{population}}\times \frac{G_{\mathrm{GDP}}}{P_{\mathrm{population}}}\times \frac{E_{\mathrm{energy}\mathrm{consumption}}}{G_{\mathrm{GDP}}}\times \frac{F_{CO2\;\mathrm{emissions}\mathrm{from}\mathrm{human}\mathrm{sources}}}{E_{\mathrm{energy}\mathrm{consumption}}}$$

The Kaya Identity has been widely applied in the energy, environmental and economic fields by the practitioners as it is clearly-structured and easy to use. For instance, it was used as a key element in the development of future emissions scenarios, as demonstrated by the Intergovernmental Panel on Climate Change (IPCC) Special Report on Emissions Scenarios in 2000 [6].

On the academic research side, there have been numerous applications of the Kaya Identity in analyzing the global, country, and regional patterns of driving forces of GHG emissions. The related studies have been summarized in Table 1.

### 2.2. Factors That Affect GHG Emissions

Demographic Factors: Demographics are often seen as a key factor influencing GHG emissions. In Kaya (1989) Identity, population growth is used as a basis to estimate GHG emissions [3]. However, Tavakoli pointed out that there is a significant difference in carbon emissions per capita, with the commission from the richest 1% of people being 175 times that of the poorest 10% [14]. This reveals that using population numbers alone to predict carbon emissions is not sufficient. Some researchers have investigated the diversity factors that influence carbon emissions per capita, such as the population age, education level, employment rate, occupation, labor productivity, and the urbanization effect [12,21,23,24,26].Economics Factors: In the Kaya Identity, GDP and GDP per capita are used to calculate the effect of economic factors on GHG emissions. Many researchers have focused on extending this equation to economic factors. Some researchers have revealed that almost all countries see industrial transformation as an important part of a low-carbon economy, as the primary, secondary and tertiary sectors have different impacts on carbon emissions and should be viewed separately, for example, tourism activity [25], economic structures [7,13,21,23,24,25]. Due to the growing gap between the rich and poor, influenced by urbanization, industrialization, and global trade, many researchers have focused on exploring alternative economic factors instead of GDP or GDP per capita, such as income variance [12], purchasing power parity [15], investment efficiency effect [21], land economic output, and land urbanization [23]. Furthermore, the market exchange rate was used as an important part of the economic analysis in Raupach, et al., [15] using Kaya Identity for global carbon emissions.Energy-related Factors: Kaya (1989) defined energy-related factors as the energy intensity of GDP (E/G) and energy emissions (F/E) [3]. However, with the technological innovations, researchers have attempted to analyze the decomposition of energy-related carbon emissions. In the area of energy system efficiency research, many researchers have focused on dividing fossil fuel energy consumption and total energy consumption (coal, peat, oil and gas), attributable to fuel quality and technology [8,9,11,12,13,16,17,18,19,20,21,23,24]. In the area of sustainable energy research, there is much research focused on renewable energy, the substitution, and fuel switching of fossil fuel types, for example, biomass, and the penetration of carbon-free energy penetration [8,10,11,16,21,22].

### 2.3. How Does Carbon Emissions Trading System/Policy Affect Carbon Emissions?

ETS has been widely used in many countries as an effective approach to mitigate GHG emissions as an incentivizing instrument as well as a market-driven policy. In 2005, the first large-scale EU ETS was established with all 15 member states of the European Union participating, and Norway, Iceland, and Liechtenstein joined the EU ETS in 2008. Furthermore, New Zealand, India, Korea, and China officially launched their national ETS in 2008, 2014, 2015, and 2021, respectively. It is worth noting that in the US, companies were able to trade carbon emissions on the Chicago Climate Exchange instead of a national ETS. Some research has investigated the effect of Emissions Trading System (ETS), mainly focusing on the EU ETS and CCETE in China. 

Regarding the case of how ETS contributes to carbon emissions, some studies have revealed the effectiveness of ETS in reducing GHG emissions. For example, a 2–5% reduction in CO_2_ emissions between 2005 and 2007 [27] and the reduction of over 1 billion tons of CO_2_ in EU countries between 2008 and 2016 [28]. In Germany, the ETS has significantly reduced carbon emissions by improving energy efficiency [29,30]. In China, the Green Credit policy’s pressure to reduce energy intensity has had a variety of impacts on different types of businesses. It has resulted in the reduction of fossil energy consumption [31], capital renewal [32], and promotion of low-carbon technology innovation [33].

Some researchers have argued that the ETS has a limited ability to curb carbon emissions and scholars have tried to find the reasons behind this. Firstly, the ETS has a limited innovation impact and cannot provide sufficient incentives for companies to create low-carbon technologies [27,34,35]. For example, Rogge et al. revealed that ETS contributes to the development of large-scale coal power generation technology in Germany, but has a limited impact on low- or zero-carbon mitigation options [34]. In China, ETS helps to reduce productive carbon emissions but does nothing for consumer emissions [36]. Secondly, some researchers are concerned about the “pollution haven” effect, which aggravates the imbalance of emissions transfers between developing countries and developed countries [37,38,39].

At the moment, the effectiveness of ETS on GHG emissions has not yet come to a consistent conclusion. Therefore, in this study, we will explore further how and why ETS is effective in reducing carbon emissions. 

### 2.4. Does Low-Carbon Economy Strategy Matter?

Climate change has emerged worldwide, with a focus on implementing a low-carbon economy to curb carbon emissions from demonstration activities. Many countries have adopted relevant strategies and policies for low-carbon economy activities. For example, European countries such as the UK, France, and Germany have set long-term targets to reduce GHG emissions by 60%, 75%, and 80%, respectively, by 2050 [40]. Many countries have realized the importance of a low-carbon economy and have tried to leverage different policy mechanisms and programs including institutional and legal frameworks (e.g., bans on single-use plastic), economic instruments (subsidies, taxes), and technology innovation. In the tax research area, for example, transport fuel taxes, energy, and carbon taxes have been widely used and recognized in many countries, such as European countries, the US, China, Kenya, Ethiopia, Costa Rica, and Turkey [41,42,43]. However, many researchers have pointed out that this policy will further increase income inequality and make it more difficult for low-income households to afford the high price of gas and electricity [44,45].

There have been various low-carbon economy strategies applied in China, and some researchers have revealed that they typically fall into two categories.

On the energy-related research side, many researchers have emphasized the importance of improving energy efficiency. Some have used the LMDI method to decompose changes in China’s GHG emissions and have shown that the reduction in energy intensity can effectively reduce carbon emissions. Examples of such studies include Fan and Ying (2007), Auffhammer and Carson (2008), Zhang YG (2009), Zhang et al. (2014), Zhang and Da (2015), Wang et al. (2016), and Naminse and Zhuang (2018) [46,47,48,49,50,51,52]. With technological innovation, the development of clean energy, and the adjustment of energy structures, effective control of GHG emissions is also possible. Examples of such studies include Wang et al. (2011), Lu et al. (2013), Dogan and Seker (2016), Zhang et al. (2020), and Xu, Le, et al. (2021) [53,54,55,56,57]. In Fujian, China, where nuclear power plays a significant role in the energy mix, it has helped to achieve China’s 2020 target of a 40–45% reduction in CO_2_ emissions intensity per unit of GDP compared to 2005, reaching around 15% by 2020 [58].

On the GDP-related research area, many researchers have found that in coal-rich regions, the intensity of carbon emissions per unit of GDP is usually reduced by growing the economy [59,60,61,62]. Zhou et al. found that in Shanxi, Anhui, and Yunnan regions, there is still a high resource intensity and negative carbon emissions footprint, which is a manifestation of the environmental burden [63].

### 2.5. Sustainable Urbanization

An increasing body of studies has focused on the effect of CCETE (ETS pilots districts and ETS pilots) on sustainable urbanization, and the literature can be divided into two categories: urban economic environment and urban green innovation.

The former focuses on the synergistic interactions between the economy and the environment systems. Firstly, some recent studies employing the Propensity Score Matching-Difference-In-Differences (PSM-DID) approach have evaluated urban carbon reduction. The results indicate that CCETE has achieved a significant reduction in ETS pilot cities’ carbon intensity [64,65,66,67,68]. For example, the energy intensity of Hubei ETS has decreased by 4.89% between 2014 and 2017 [67]. According to Tang’s study, the average urban carbon intensity of cities covered by the pilot is 80% lower than others. Furthermore, some researchers are concerned about the negative economic impact of ETS. For example, in 2014, the first year of trading in Hubei ETS, Hubei’s GDP fell by about 0.06%, or about $1.48 billion, and 88.09% of the industries involved in the transaction were negatively affected [69]. However, recent studies have argued that the negative impact is limited and still positive for sustainable urbanization [66,67,68,69]. 

The second category analyzes the effect of ETS on urban innovation. According to the “Porter Hypothesis”, technology innovation encouraged by environmental regulation policy is an important pathway to achieve sustainable development [70]. Liu et al. (2020) analyzed the innovation index and found that ETS pilots have significantly spurred the technology innovation of ETS pilot cities [71]. Li et al. (2021) used a dataset of green patent applications to measure the level of urban green innovation, and the results indicate that ETS has a positive impact on green innovation activities [72].

## 3. Policy Review and Analysis

### 3.1. Sustainability Policy Framework Development in China, from Policy-Oriented to Market-Oriented

China realized the importance of energy conservation and emissions reduction and promulgated the Report on Strengthening Energy Conservation in 1980, which marked the first time that energy conservation was included in the country’s macro management. Between 1980 and 1992, several mandates were proposed and formulated, resulting in a comprehensive regulatory system based on energy efficiency management. As China was in the planned economy phase during this period, all economic activities depended on the directive plans issued by the government, and the energy management framework of China was policy-led as well. 

The energy policy can be divided into two parts: energy conservation and pollution prevention. Three Energy Conservation Policies were passed in 1980, 1984, and 1996. A series of policies and regulations in the “Interim Regulations on Energy Conservation Management” advanced the development of a regulatory system for energy efficiency management. At the same time, a supporting legal system in environmental legislation and pollution prevention began to take shape. The management of atmospheric, sewage, and waste discharges has continuously improved, including the setting of discharge standards, the establishment of discharge permits, and the collection of discharge charges.

China has a tracked record of adjusting its energy structure and developing a sustainable energy strategy framework, with a large number of policies related to new energy sources. In 1995, China introduced the “Outline for the Development of New and Renewable Energy Sources 1996–2010” and shifted the focus towards a sustainable industrial structure and energy consumption structure, such as by encouraging the adoption of clean and/or renewable energy sources such as wind, solar, and geothermal energy. In 1997, the Law of the People’s Republic of China on Energy Conservation was passed, with a focus on tapping into the market benefits of energy conservation and providing legal safeguards for energy conservation. This was followed by the implementation of the first medium and long-term energy development plan outline (2004–2020) in 2004 and the adoption of the Renewable Energy Law in 2006 to support renewable energy.

The government also started to focus on market mechanisms to limit the carbon emissions, including the imposition of emissions charges and fixed tariffs, as mentioned in the 2003 Regulation on the Collection and Use of Emissions Charges. The collection of sewage charges from enterprises is not only an effort to combat pollution, but also to reduce the financial burden on the government in terms of environmental management.

In the past decade, China has adopted a market-based sustainable energy strategy framework by developing low-carbon technologies and implementing low-carbon pilot cities. Localized carbon taxes, carbon funds, preferential sustainable energy prices and green credits have been established in low-carbon pilot cities. The first batch of low-carbon pilot provinces and cities include five provinces (Guangdong, Liaoning, Hubei, Shaanxi, and Yunnan) and eight cities (Tianjin, Chongqing, Shenzhen, Xiamen, Hangzhou, Nanchang, Guiyang, and Baoding).

For energy-intensive enterprises, resource taxes on crude oil, natural gas, and coal were raised in China’s 12th Five-Year Plan. For green enterprises, the Energy Efficiency Credit Guidelines provides preferential loan prices and the power to guarantee loan targets on a priority basis. The sustainable energy strategy framework has optimized energy intensity through technology development, and substitution between different types of fossil fuels has effectively controlled carbon emissions. The fossil fuel substitution effect has become an essential part of the process of building low-carbon development.

### 3.2. Carbon Emissions Trading System (ETS)

Since 2011, the China Carbon Emissions Trade Exchange (CCETE) has been established with seven pilot districts and eight ETS pilots. In 2011, Beijing, Shanghai, Tianjin, Guangdong, Shenzhen, Hubei, and Chongqing were selected as the pilot districts by the National Development and Reform Commission (NDRC).

Then, China started to launch eight ETS pilots in Beijing, Tianjin, Shanghai, Fujian, Hubei, Shenzhen, Guangzhou, and Chongqing. In 2013, the Shenzhen ETS made the first carbon credit transaction, which marked the beginning of carbon trading pilot city operations. In the ETS pilots, only enterprises whose comprehensive energy consumption meets specific standards are eligible for emissions allowance trading. For example, according to “the Interim Measures for Carbon Emissions Rights Management and Trading in Hubei Province”, promulgated in April 2014, a total of 138 enterprises are eligible for carbon trading in the Hubei Carbon Trading Pilot.

The establishment of pilot districts and ETS pilots attempts to contribute to the construction of a national carbon emissions trading market. In 2017, the central government completed the national carbon emissions trading market. According to the “National Carbon Emissions Trading Management Measures (for Trial Implementation)” promulgated in November 2020, CCETE covers more than 20 industries such as electricity, steel, and cement, with nearly 3000 key emissions units, making it the second largest carbon market in the world. The Chinese national ETS was officially launched in July 2021.

### 3.3. Sustainable Energy Strategy

As mentioned in the policy development timeline, the sustainable energy strategy has been promoted in China for decades. It can be defined as the strategy in using different types of energy resources. The sustainable energy resources include hydro, nuclear, wind, and solar power, and the electricity power created from the sustainable energy resources is considered as the adoption of a sustainable energy strategy. On the other hand, the electricity power that is created from the fossil fuel is seen as the use of the unsustainable energy strategy. We follow [73] to use to the fossil fuel substitution effect (SUB) to represent the quantified level of the adoption of a sustainable energy strategy. The calculation of SUB is expressed as
(2)$$SUB=\frac{electricity\;power\;created\;by\;sustainable\;energy\;resources}{electricity\;power\;created\;by\;all\;types\;of\;energy\;resources}$$

SUB is one of the important indicators of the low-carbon economy transition, sustainable energy adoption, and technology development. With the development of the sustainable energy strategy, SUB has been increasing across the country from 2009 to 2019 with some geographical differences. Figure 3 shows that after 2014, the central and western regions of China, mainly Qinghai, Sichuan and Hubei, have made extensive use of new energy generation, mainly hydro, to replace traditional thermal power generation, such as the famous Three Gorges Dam in Hubei Province. The SUB values for these regions have been much higher than those of other regions, at 73.11%, 67.48%, and 66.33%, respectively, in 2009. In contrast, most of China’s regions have low SUB values, with 16 provinces having SUB values of less than 10%. Shanghai, Tianjin, Jiangsu, Beijing, and Hebei, which are more urbanized, have lower new energy usage rates, with SUB values of 0.00%, 0.00%, 0.08%, 0.15%, and 0.33%, respectively. Meanwhile, the SUB values for resource-dependent cities are also low, for example 0.66% in Inner Mongolia and 2.36% in Heilongjiang. This means that the eastern and northern regions of China are facing a greater challenge of energy transition and industrial structure transformation. After ten years of continuous efforts, in 2019, only three provinces and cities had SUB values below 10%. This means that Chinese provinces and cities have completed their basic energy transition and reduced their dependence on traditional fossil energy sources. The province with the biggest change during this decade is Yunnan Province, which has improved from 53.19% to 90.16%, essentially freeing itself from dependence on traditional energy sources.

### 3.4. The Carbon Emissions Reduction Effects from Carbon Emissions Trading System and Sustainable Energy Strategy across China

Hypothetically, the SUB shall be negatively associated with the carbon emissions, according to CERGC. Figure 4, Figure 5, Figure 6, Figure 7 and Figure 8 demonstrate the negative correlation between carbon emissions per capita (CEC), carbon emissions per unit of energy consumption (CEEC), carbon emissions per capita of real GDP (CERGC), and various carbon dioxide calculation models and SUB. Most provinces show a clear trend of decreasing carbon emissions with increasing SUB values. The most pronounced downward trend is seen in CERGC, with decreases ranging from 7.78% to 202.26%. This is due to the greater use of carbon emissions per unit of GDP as a measure at the policy level. Shanxi, Henan, Guizhou, Hebei, Shandong, and Sichuan are the provinces with the largest changes, with CERGC decreasing by 202.26%, 153.60%, 143.91%, 107.93%, 106.35%, and 104.44%, respectively. Meanwhile, with technological innovation and the promotion of alternative energy sources, CEEC showed a slow downward trend. In 2019, most provinces had a CEEC below 3%. In provinces with a high proportion of primary and secondary industries, the CEEC is relatively high, for example, Shanxi and Shaanxi, with a CEEC of 8.15% and 4.54%, respectively. Finally, the CEC does not show a clear downward trend, with only eight provinces showing a downward trend, namely Beijing, Jilin, Henan, Chongqing, Sichuan, Yunnan, Shaanxi, and Gansu.

SUB implies the geographical inequalities that are caused by the unbalanced degree of urbanization and the distribution of resources. This feature is evident on the map, with the east and west showing very different states of SUB development. This inequality is mitigated by the operation of the CCETE, which includes Tianjin, Beijing, Shanghai, Shenzhen, and Guangdong, all of which are located in provinces with high levels of urbanization, low SUB values, and dependence on traditional energy sources. According to Figure 9, we can clearly see that the establishment of these CCETE pilots has effectively promoted the development of new energy uses in their regions. In Tianjin and Beijing, for example, carbon emissions trading has been introduced since 2013, gradually driving the adoption of new sustainable energy resources within the area. In 2012, Tianjin’s SUB was 0.00%, but in 2013 it increased to 3.6%, achieving a breakthrough from nothing to something. Hebei’s SUB increased from 0.33% in 2009 to 15.48% due to its proximity to both Beijing and Tianjin. It indicates that there is synergy between the two measures in terms of carbon emissions reduction.

The synergy between the sustainable energy strategy and ETS motivates us to investigate further their impacts on carbon emissions reduction. Unlike the previous studies, which focused on either ETS or a sustainable energy strategy, this study explores the quantitative contribution of the two measures together, along with other factors which are proven to be closely associated with carbon emissions, such as population, GDP, and energy consumption.

## 4. Data and Methodology

### 4.1. Data

This study uses a panel dataset consisting of 31 provinces in China from 2009 to 2019, and Tibet Province is not included in the dataset due to missing data. Standardized carbon emissions data are from Carbon Emissions Accounts & Datasets (CEADs) [74,75]. Control variables include population, policy strategy, economic factors, and energy-related factors. The data are collected from the China Urban Statistical Yearbook and the China Energy Statistical Yearbook for 2010–2020. In our study, the population is represented by the population in each province; economic activities are presented by GDP per capita in each province; and energy-related factors are represented by the total energy consumption in each province (million tons of standard coal). The policy strategy factors are presented by two variables: ETS and SUB. ETS is a dummy variable based on whether CCETE is established in each province at the time, and the fossil fuel substitution effect (SUB) is used to show what extent the fossil fuel is substituted by other sustainable energy resources. We follow Zeng and Tong (2017) to calculate the SUB of each province in each year [73]. As defined in last section, the fossil fuel substitution effect (SUB) represents the substitution effect of new energy sources for traditional energy sources.

Table 2 shows a brief description and statistical summary of the variables used in the empirical test.

### 4.2. Empirical Model

The OLS regression method is used to explore various emerging factors that are possibly affecting regional carbon emissions. Derived from the KAYA model (1989) that is shown in Equation (1), we construct the empirical model (Equation (3)) to test how the measures, namely the sustainable energy strategy and carbon trading system, have affected the carbon emissions in the area.
(3)$${CO2}_{i,t}=\beta_{0}+\beta_{1}\times {SUB}_{i,t}+\beta_{2}\times {ETS\_N}_{i,t}+\beta_{3}\times {ETS\_R}_{i,t}+\beta_{4}\times {GDP}_{i,t}+\beta_{5}\times {POP}_{i,t}+\beta_{6}\times {ENG}_{i,t}+\varepsilon_{i,t}$$
where ${CO2}_{i,t}$ denotes the CO_2_ emissions in province *i* at year *t*; ${SUB}_{i,t},{GDP}_{i,t},{POP}_{i,t}\;\mathrm{and}\;{ENG}_{i,t}$ represent the fossil fuel substitute effect, real GDP, population, and the total energy consumption in province *i* at year *t*, respectively. ${ETS\_N}_{i,t}\;\mathrm{and}\;{ETS\_R}_{i,t}$ are the dummy variables: ${ETS\_N}_{i,t}$ is equal to 1 if there was a carbon trading policy anywhere across the country in year *t*; and equal to 0 otherwise; ${ETS\_R}_{i,t}$ is equal to 1 if there was a carbon trading policy in province *i* at the year *t*; and equal to 0 otherwise. The *ETS_N* gives the signal of potential financial incentives from carbon emissions trading. Previous studies suggest that the firms prefer to trade carbon emissions credit locally (within the region); therefore, we hope to investigate the effectiveness of the nationwide signal as well as the local pilot practice, which is represented by *ETS_R* [76,77,78]. Therefore, in our empirical test, three sub-panels of regression analysis are conducted.

### 4.3. Stationarity Check: Unit Root Tests

We use the ADF (Augmented Dickey Fuller) test and Phillips–Perron (PP) test to test the stationarity of the variables in this paper. Table 3 presents the unit root tests for all variables with a null hypothesis for which there is a unit root.

The variables of real GDP, population, and fossil fuel substitution effect show that they are not stationary; thus, first-differenced data series are used in the empirical test. The variables of total energy consumption and CO_2_ emissions data have no unit root, and therefore will not convert to be stationary by taking the first difference. 

## 5. Results and Discussion

In this section, we run the empirical test with three sub-panels of data. The first panel explores the impacts of all the explanatory variables on CO_2_ emissions in the area. In panel 2, we assume that the impact of the carbon trading policy at the national level could have been shared among all provinces; thus, we explore all the explanatory variables’ impacts except the carbon trading policy across the country. In panel 3, we assume that the impact of the regional level carbon trading policy could have been omitted by the announcement of the national level carbon trading policy. Thus, we investigate the impact of all explanatory variables except the carbon trading policy at the regional level. The empirical estimation results of the three sub-panel regression models are presented in Table 4. 

As demonstrated in Table 4, across all three groups of tests, the fossil fuel substitution effect is significantly associated with the local CO_2_ emissions in a reverse way. This indicates that the more fossil fuel is substituted by sustainable energy resources in the area, the lower the regional CO_2_ emissions. In our study, the fossil fuel substitution effect is used to represent the sustainable energy resource strategy in each province. The empirical result shows that when a province adopts a higher level of substituted sustainable energy resources, CO_2_ emissions can be reduced if other variables stay equal. 

Meanwhile, total energy consumption is found to have a positive effect on CO_2_ emissions in the area, which is consistent the previous findings based on Kaya’s (1989) Model. This model suggests that the more energy is consumed, the higher the level of CO_2_ emissions. 

It is interesting to discuss what we have discovered about the carbon emissions trading system. As we separate the policy into two levels—the national level and the regional level—as two variables, the results show that the regional level of carbon trading policy has negative impacts on CO_2_ emissions, while the national level of carbon trading policy only has statistically significant negative impacts on CO_2_ emissions in the areas where we do not include the regional level of the carbon emissions trading system. This implies that provinces that have implemented a carbon emissions trading system are inclined to reduce their CO_2_ emissions for the financial rewards via trading, while provinces without a carbon emissions trading system have no such incentive to reduce their CO_2_ emissions. This might be caused by the hurdles of cross-province trading for carbon credit, as suggested in previous studies. The national-level carbon emissions trading system is only effective when the regional-level carbon emissions trading system is not included in the model. This suggests that the national-level carbon trading policy varied across different areas in the country, and that it is effective as a signal of potential financial rewards from the future carbon credit trading. Without the pilot practice of a regional level carbon emissions trading system, the incentives from such signals are limited.

In regard to the economic factor, the real GDP has a negative impact on CO_2_ emissions in the area which shows that provinces with highly productive industries for the local economy have the capability to reduce their CO_2_ emissions. However, the provinces that rely on the energy-dependent industries for the local economy struggled to reduce CO_2_ emissions in the area. The demographics and population are found to be positively affecting the CO_2_ emissions in the area, which is in line with previous findings based on Kaya’s (1989) Model.

## 6. Conclusions and Policy Implications

Both the carbon trading system and sustainable energy strategy have been widely discussed over the past decade, but only a handful of studies have focused on measuring the effectiveness of both. This study uses the panel regression to investigate the impact of both carbon emissions trading policies and the ongoing sustainable energy strategies across 30 provinces in China. The results show that sustainable energy strategies are effective in reducing carbon emissions within the province. The sustainable energy strategy varies across different provinces and over the past decades, so it is important to review the various strategies, and the findings of this study confirm the effectiveness of such strategies in reducing carbon emissions. The finding that substituting fossil fuels with sustainable energy resources can effectively reduce carbon emissions in provinces where high-level sustainable energy strategies are adopted has several policy implications. Firstly, it suggests that governments should prioritize the development and implementation of sustainable energy strategies to reduce carbon emissions. This could involve investing in renewable energy sources such as solar, wind, and hydropower, as well as implementing policies that promote energy efficiency and reduce energy consumption. Secondly, the finding highlights the importance of local factors in determining the effectiveness of sustainable energy strategies. Policymakers may need to consider the unique characteristics of different regions when developing and implementing sustainable energy policies. For example, urban areas may require different policies than rural areas, as they often have higher levels of energy consumption and greater dependence on fossil fuels.

Our finding suggests that urban areas may be particularly well-suited to sustainable energy strategies. Cities often have the infrastructure and resources necessary to support renewable energy projects, such as access to financing, skilled labor, and technological innovation. Additionally, urban areas may benefit from the economic and environmental benefits of sustainable energy, such as reduced air pollution and improved public health.

To fully realize the potential of sustainable energy in urban areas, policymakers may need to adopt a comprehensive approach that addresses the unique challenges of urbanization. This could involve promoting sustainable transportation options, such as electric vehicles and public transit, as well as investing in energy-efficient building design and the development of smart grids that can better manage energy consumption. Overall, the finding suggests that sustainable energy strategies can be an effective way to reduce carbon emissions in the context of urbanization, but that these strategies must be tailored to the specific needs and characteristics of different regions.

Meanwhile, the findings also suggest that the national level of carbon emissions trading policy or announcement has little impact on carbon emissions change, while the regional level carbon emissions trading system set-up helps to reduce carbon emissions locally. This might be because of the current nature of carbon credit trading: most players prefer to trade the carbon credit locally rather than outside the region. Therefore, having a carbon credit trading policy in place at each regional level will be helpful in reducing the carbon emissions locally. The finding that the carbon emissions trading system (ETS) only reduced carbon emissions in the provinces where the ETS pilot was implemented has several policy implications. First, it suggests that a more comprehensive approach is needed to reduce carbon emissions across the entire country, rather than just in specific regions. This could involve implementing a nationwide ETS or other policies that target emissions reduction in areas that are not covered by the pilot program. Secondly, the finding also suggests that the effectiveness of the ETS may depend on local factors, such as the level of industrial development and the availability of alternative energy sources. Policymakers may need to consider these factors when designing and implementing carbon reduction policies in different regions.

Our findings highlight the need to address the unique challenges of reducing emissions in urban areas, where there are often higher levels of energy consumption and a greater dependence on fossil fuels. Urbanization may also increase the demand for energy intensive infrastructure, such as buildings and transportation systems, which could further contribute to carbon emissions. To address these challenges, policymakers may need to develop tailored policies and strategies that address the specific needs and characteristics of urban areas. This could involve promoting renewable energy sources, encouraging energy efficient building design, and incentivizing the use of low-carbon transportation options. Overall, the finding highlights the need for a comprehensive and localized approach to reducing carbon emissions in the context of urbanization.

In recent decades, China has undergone rapid urbanization and development. It is crucial to pay close attention to the environmental impacts resulting from urbanization. The fast-growing economy in China has also led to high energy consumption and thus higher carbon emissions. These environmental issues can be effectively addressed by adopting sustainable energy strategies and implementing a carbon trading system. A well-designed and actively-traded carbon emissions trading system can be of great help in reducing carbon emissions. The good practices in the pilot provinces in China can be a useful showcase for countries around the world that are considering setting up their own carbon emissions trading systems. 

There are a few limitations to this study. In terms of sustainable energy strategy, we can only use the fossil fuel substitution effect as a proxy to measure it quantitatively. As the variety of strategies continues to grow and quantified indicators emerge in the future, more precise measures of sustainable energy strategy can be employed in empirical studies. In addition, the carbon emissions data are at the province level, corroborating findings from the literature, but these data may not be adequate to present all the carbon emissions information within a province. Future research can examine carbon emissions reduction in different sectors to address this issue. Lastly, the synergy effect between two levels of ETS policy is not explored sufficiently in this study due to the data limitations. We use a dummy variable to represent the ETS on national and regional levels, which cannot reflect the development of the ETS pilot over time. Future research can examine the degree of development of ETS and its impact on carbon emissions reduction.

## Figures and Tables

**Figure 1 ijerph-20-05549-f001:**
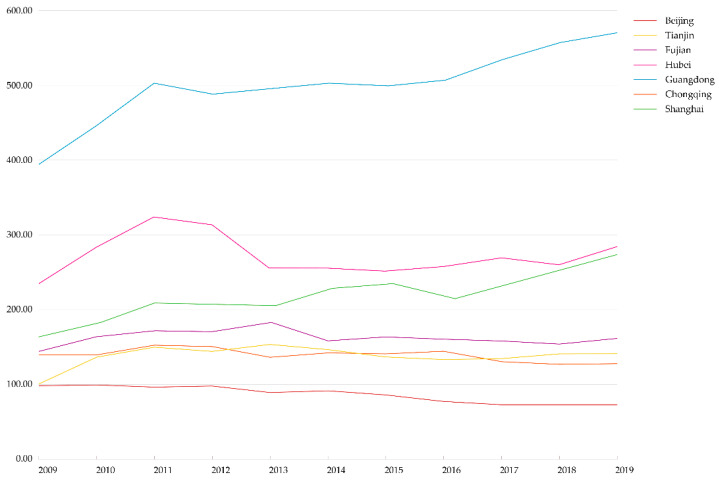
The Carbon Emissions across 7 CCETE provinces in China (2009–2019).

**Figure 2 ijerph-20-05549-f002:**
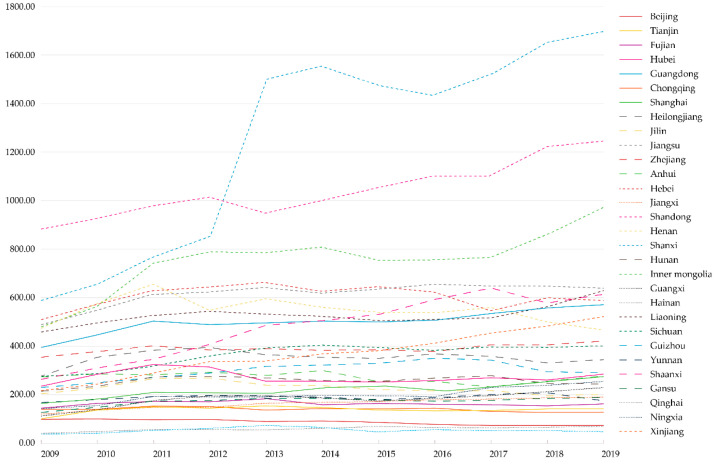
The Carbon Emissions across 30 Provinces in China (2009–2019).

**Figure 3 ijerph-20-05549-f003:**
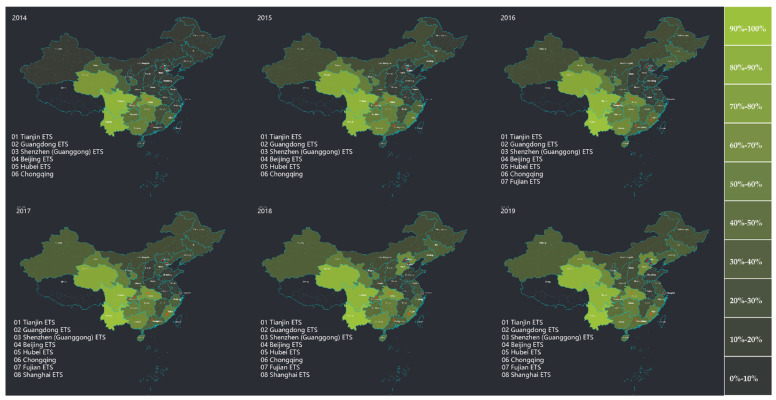
The Sustainable Energy Strategy (SUB) change across 30 selected provinces in China (2014–2019).

**Figure 4 ijerph-20-05549-f004:**
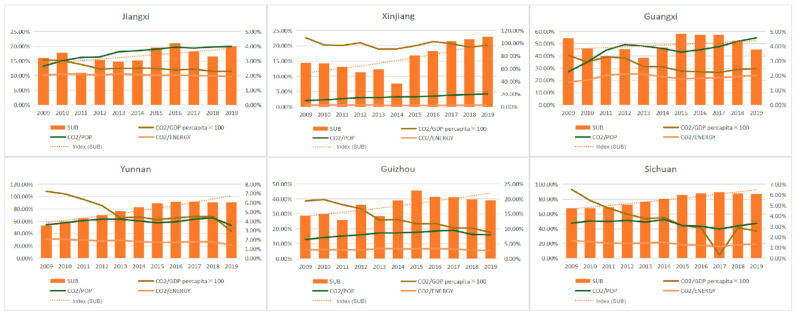
The Carbon Emission and SUB change across 30 provinces in China (2009–2019).

**Figure 5 ijerph-20-05549-f005:**
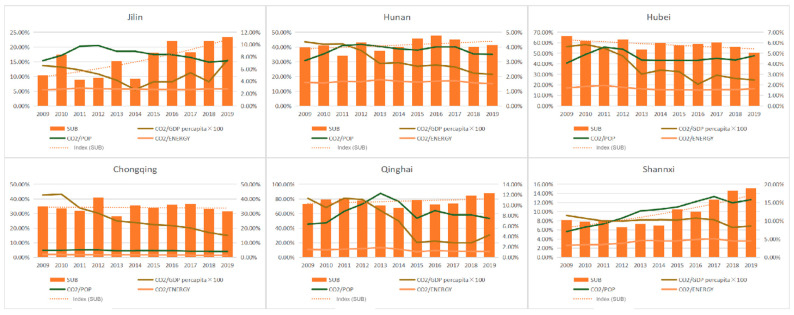
The Carbon Emission and SUB change across 30 provinces in China (2009–2019).

**Figure 6 ijerph-20-05549-f006:**
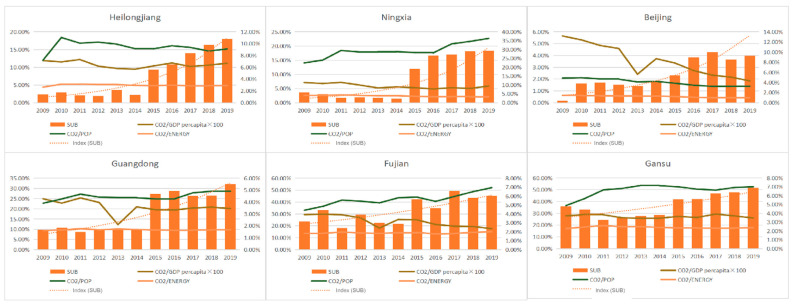
The Carbon Emission and SUB change across 30 provinces in China (2009–2019).

**Figure 7 ijerph-20-05549-f007:**
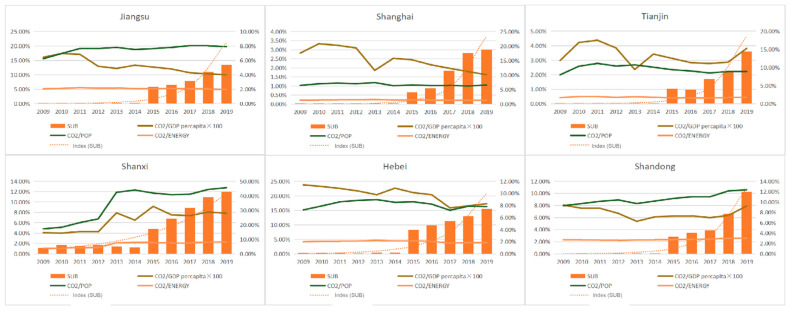
The Carbon Emission and SUB change across 30 provinces in China (2009–2019).

**Figure 8 ijerph-20-05549-f008:**
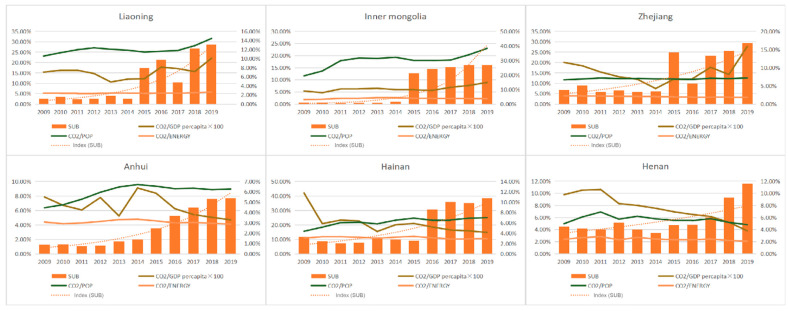
The Carbon Emissions and SUB change across 30 provinces in China (2009–2019).

**Figure 9 ijerph-20-05549-f009:**
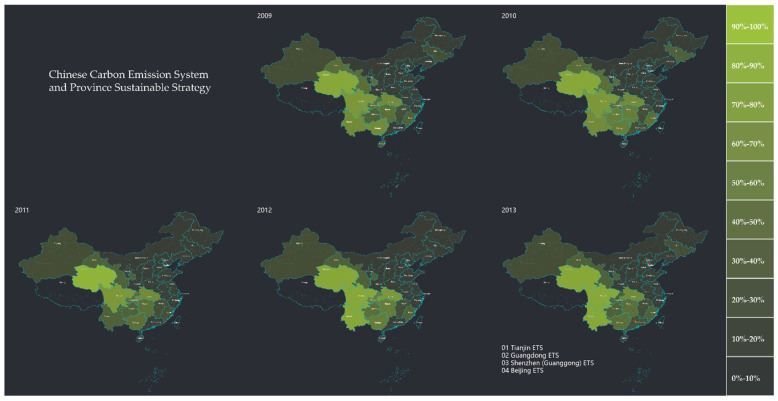
The SUB change across selected 30 provinces in China (2009–2011).

**Table 1 ijerph-20-05549-t001:** Summaries of the previous empirical investigations on GHG emissions based on Kaya Identity.

Citation	Country	Time Period	Demographic Factors (P)	Economics Factors	Energy Related Factors
Ma and Cai, 2018 [7]	China	2001–2015	N.A.	5	N.A.
Mahony, 2013 [8]	Ireland	1990–2010	Population	GDP	E = Total Primary Energy Requirement (TPER) of all fuel types; FFi = TPER of fossil fuel type i; FF = TPER of all fossil fuels; C = Total CO_2_ emissions from all fossil fuel types; Ci = CO_2_ emissions from fossil fuel type i
Jung et al., 2012 [9]	Korea	2002–2009	Population	GRDP is the gross regional domestic product	TEC is total energy consumption; EC is fossil fuel energy consumption.
Ma and Stern, 2008 [10]	China	1971–2003	Population	GDP	Carbon emissions from fossil fuels consumption; fossil fuels consumption (coal + oil + natural gas); carbon-based fuel consumption (FF + biomass); total fuels consumption (CF + carbon-free fuels)
Wang et al., 2005 [11]	China	1957–2000	Population	GDP; GDP per capita	The mean CO_2_ emissions coefficient of fossil fuels; the fuel share in total energy consumption
Chontanawat, 2019 [12]	ASEAN	1971–2013	Population in million persons	Real GDP: defined and measured at constant price in million 2005 USD	CO_2_ emissions flux in Mt of CO_2_ emissions; primary energy supply in ktoe; fossil fuel consumption in ktoe
Ortega-Ruiz et al., 2020 [13]	India	1999–2016	Population	Economic activity (act), economic structure (str)	Energy intensity (int), and energy mix (mix)
Tavakoli, 2018 [14]	Top ten emitter country of 2015 (China, United States, India, Russian Federation, Japan, Germany, Korea, Canada, Iran and Brazil)	1971–2011	Population	GDP capita	Energy Intensity, Carbon Intensity
Raupach, et al., 2007 [15]	U.S., China, Japan, India, the European Union (EU), the nations of the Former Soviet Union (FSU), developed (D1), developing (D2), and least-developed (D3) countries	1980–2004	Population	GDP	Energy consumption, regional intensities
Eskander, Nitschke, 2021 [16]	UK	2012–2019	Population	Income	Total energy consumption of all fuel types. Total CO_2_ emissions
Lin and Raza, 2019 [17]	Pakistan	1978–2017	Population	GDP	CO_2_ emissions, energy utilization, fossil fuel emissions
Wang et al., 2020 [18]	US	1997–2016	Population	GDP	Total CO_2_ emissions, CO_2_ emissions of fuel type in sector, the energy needs by fuel type in sector, the total energy needs and industrial added value in sector
Okorie, 2021 [19]	Nigeria	1960–2019	Population	GDP	Energy Structure, proxies for some of the factors, input factor, i.e., energy consumption, output factor, i.e., carbon emissions
Yang et al., 2020 [20]	China	2000–2015	Population	GDP	Carbon intensity
Pui, Othman, 2019 [21]	Malaysia	2000–2016	The population-to-employment effect, The urbanization effect, The per capita CO_2_ effect	Scale effect, economic structure effect, the capital-labor substitution effect.	The sum of energy intensity effect; the renewable energy penetration effect; the fossil fuels-renewable energy substitution effect; the investment efficiency effect
Lima et al., 2016 [22]	Latvian and Lithuanian	1995–2019	Population	GDP	CO_2_ emissions, CO_2_ emissions from fossil fuel type
Wang et al., 2014 [23]	China	1995–2011	Population	Land economic output, land urbanization, urban area of per capita, population urbanization	Energy mix, energy intensity, industrial structure
Xu et al., 2014 [24]	China	1995–2010	Population	Economic output	Energy structure, energy intensity, industry structure
Robaina et al., 2016 [25]	Portuguese	2000–2008	N.A.	Total value added generated by tourism; the value added generated by tourism	Carbon intensity, energy mix, energy intensity, tourism consumption by the value added generated by tourism (VA effect)

**Table 2 ijerph-20-05549-t002:** Description and statistics summary of variables.

Variables	Description	Mean	SD	Max	Min
CO_2_	Total apparent CO_2_ emissions (mt)	360.33	288.66	1700.04	35.46
ETS	Equal to 1 if there is carbon emissions trading system; equal to 0 otherwise	/	/	/	/
SUB	Fossil fuel substitution effect (%)	0.25	0.24	0.92	0.00
POP	Population	1364.71	912.74	4389.73	32.5
GDP	Real GDP per capita (rmb)	587,946.91	545,055.40	7,321,626.00	31,364.00
ENG	Total Energy Consumption (104 tce)	14,583.88	8658.96	41,390.00	1232.52

**Table 3 ijerph-20-05549-t003:** Unit root test results.

Variable	Cross-Sections	Observations	ADF		PP	
			Level	1st Diff	Level	1st Diff
**GDP**	30	300	25.79	94 ***	47.21	73.82 *
**POP**	30	300	51.87	101.88 ***	84.94	263.07 ***
**SUB**	30	300	24.29	89.91 ***	52.5	194.52 ***
**ENG**	30	300	103.34 ***		252.8 ***	
**CO_2_**	30	300	85.98 *		141.16 ***	

The reported numbers represent test statistics. *** and * indicate the rejection of the null hypothesis at the 1% and 10% level of significance, respectively.

**Table 4 ijerph-20-05549-t004:** Estimation results of the panel regression models.

Dep. Var. = log(CO_2_)	Panel 1	Panel 2	Panel 3
constant	44.9099 **	35.5507 **	40.4881 **
	(17.7649)	(15.1943)	(17.2656)
**Explanatory Variables**			
**log(SUB)**	−35.4778 *	−25.4614 **	−34.2885 *
	(20.5445)	(12.0609)	(18.3942)
**ETS_N**	−15.7700		−4.3127 *
	(2.7372)		(2.2020)
**ETS_R**	−15.7700 ***	−17.0532 ***	
	(3.9135)	(3.2382)	
**log(GDP)**	−0.0013 **	−0.0005	−0.0019 ***
	(0.0006)	(0.0005)	(0.0006)
**log(POP)**	0.0183	0.0130	0.0236 *
	(0.0128)	(0.0117)	(0.0127)
**ENG**	0.0223 ***	0.0227 ***	0.0228 ***
	(0.0012)	(0.0010)	(0.0012)
no. of observations	298		
R-squared	0.9894	0.9917	0.9893
adjusted R-squared	0.9880	0.9906	0.9879
S. E. of regression	59.87	64.08	59.94
effect specification	cross-section fixed effect		

Standard errors in parentheses; *** *p* < 0.01, ** *p* < 0.05, * *p* < 0.1.

## Data Availability

The information used in the analysis is accessible from the public data sources.

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
