# Peer review of "The Roles of Carbon Trading System and Sustainable Energy Strategies in Reducing Carbon Emissions—An Empirical Study in China with Panel Data"

_ijerph, 2023, doi:10.3390/ijerph20085549_

Round 1

Reviewer 1 Report

This study explores the effectiveness of the two emerging methods to reduce carbon emission. The research is interesting, but exists many deficiencies regarding grammar and sentence. I suggest authors to invite a native English speaker for revising the manuscript.

1. I suggest authors read more academic papers. The authors’ address is not completed.

2. Abstract, the first conclusion, page 1, line 19. What’s meaning of our production and consumption? Besides, renewable/sustainable energy, they are different. So, what’s the meaning?

3. In the key words, the authors emphasize the urbanization. However, in the conclusions, the authors don’t propose the urbanization.

4. The authors don’t make strict literature review regarding emission trading rights. Some academic paper may help you complete the Literature review.

(1) Can green credit policy promote low-carbon technology innovation? This paper explains why green credit makes an influence on market.

(2) Crossing the rivers by feeling the stones: The effect of China's green credit policy on manufacturing firms' carbon emission intensity. This paper introduces what changes will enterprises make to green credit.

(3) Green finance policy coupling effect of fossil energy use rights trading and renewable energy certificates trading on low carbon economy: Taking China as an example. This paper energy use rights trading and how it work in the market.

5. The authors should restructure the whole manuscript, especially Section 2. I advise the authors change the structure into:

     1. Introduction

     2. Literature review

     3. Policy analysis

     4. Data and methodology

     5. Changes of sustainable energy strategy

6. Results and discussion

7. Conclusions and policy recommendation

6. The manuscript lacks the policy recommendation, which is the core inspiration of the research. So, authors should add the policy recommendations or policy implications.

7. References exist many Chinese journal. The authors should translate Chinese into English for the whole citation.

8. The main issue for this manuscript is the existing errors for grammar, sentence and expression. I suggest the authors invite a native English scholar to revise the manuscript.

Reviewer 2 Report

1. The quality of the figures could be improved, such as the Figure 1 to 5.

2. The methods in this paper should be presented in more detail.

3. The results and discussion are not sufficient. The discussion could combined with the background in section 2.

4. This paper has focused on the background and literature review, however, what is the important methods, results, and conclusions in this paper is not sufficient. The more details should be provided.

5. The abstracts and introduction should be improved.

6. Is the introduction includes the literature review? Is it necessary to presented the review independently?

7. Conclusions should be presented in more clear.

Reviewer 3 Report

This study explores the effectiveness of two emerging methods to reduce carbon emission, which are carbon trading system and sustainable energy strategy, in the process of urbanization. The effectiveness of the two methods with panel data across 30 provinces in China from 2009 to 2019 were investigated by reviewing the policy in the past decades. It is an interesting research which can help us to get more knowledges about the roles of Carbon Trading System and Sustainable Energy Strategies in Reducing Carbon Emission. I just have some suggestions for this manuscript. 

1) The roles of carbon emission trading system and sustainable energy strategy in reducing carbon emission are investigated in this paper. However, quantitative contribution of those two methods to CO2 emission reduction is not obtained. The conclusion “carbon emission trading policy is vital for achieving the aim of carbon emission reduction, the sustainable energy strategies is playing a pivotal role as well” is an obvious truth, which can be understand even without any calculation. After all those analyses, the author is highly suggested to dig deeper and provide more detailed results and findings.

 2) The SUB, POP, GDP and ENG are taken into consideration as control variables for CO2 emission. As mentioned by the authors, CO2 emission analysis is a complex topic. And I believe there must be other factors relating to the CO2 emission, such as energy efficiency, industry structure, and even inter-province merchandise trade, etc. How to make the results be reasonable without considering those factors? At least, the estimation errors of this modeling analysis need to be presented.

 3) What’s the different effect of regional and national policies on CO2 emission? how to distinguish them? How to consider the synergetic effect of them? More details need to be provided.

 4) some specific comments:

Please define “carbon emission” and “CO2 emission”. Are they the same or not?

Abstract, line 4, “urbanisation” should be “urbanization”.

Figure 1, the lines’ colors of different provinces are too close to each other to distinguish, such as the Shanxi, Henan and Qinghai. It can be addressed by adopting different line types. Same as Fig. 2.

Figure 3 and 4, the legends are unclear.

Figure 5, it is too small to see clearly.

Round 2

Reviewer 1 Report

The manuscript is improved a lot. You should continue to improve the format, such as the References.

Reviewer 2 Report

The paper was revised well.

Reviewer 3 Report

Well done, my questions have been addressed. Your paper is worth to be published.